# Knobs, Adhesion, and Severe Falciparum Malaria

**DOI:** 10.3390/tropicalmed8070353

**Published:** 2023-07-04

**Authors:** Mark F. Wiser

**Affiliations:** Department of Tropical Medicine and Infectious Disease, Tulane University School of Public Health and Tropical Medicine, 1440 Canal Street, New Orleans, LA 70112, USA; wiser@tulane.edu; Tel.: +1-504-988-2507

**Keywords:** malaria, *Plasmodium falciparum*, cytoadherence, sequestration, endothelium, KAHRP, PfEMP1, knobs, antigenic variation

## Abstract

*Plasmodium falciparum* can cause a severe disease with high mortality. A major factor contributing to the increased virulence of *P. falciparum*, as compared to other human malarial parasites, is the sequestration of infected erythrocytes in the capillary beds of organs and tissues. This sequestration is due to the cytoadherence of infected erythrocytes to endothelial cells. Cytoadherence is primarily mediated by a parasite protein expressed on the surface of the infected erythrocyte called *P. falciparum* erythrocyte membrane protein-1 (PfEMP1). PfEMP1 is embedded in electron-dense protuberances on the surface of the infected erythrocytes called knobs. These knobs are assembled on the erythrocyte membrane via exported parasite proteins, and the knobs function as focal points for the cytoadherence of infected erythrocytes to endothelial cells. PfEMP1 is a member of the var gene family, and there are approximately 60 antigenically distinct PfEMP1 alleles per parasite genome. Var gene expression exhibits allelic exclusion, with only a single allele being expressed by an individual parasite. This results in sequential waves of antigenically distinct infected erythrocytes and this antigenic variation allows the parasite to establish long-term chronic infections. A wide range of endothelial cell receptors can bind to the various PfEMP1 alleles, and thus, antigenic variation also results in a change in the cytoadherence phenotype. The cytoadherence phenotype may result in infected erythrocytes sequestering in different tissues and this difference in sequestration may explain the wide range of possible clinical manifestations associated with severe falciparum malaria.

## 1. Introduction

Malaria is caused by protozoan parasites of the genus *Plasmodium* and is transmitted by mosquitoes. The parasite has a complex life cycle involving specialized invasive forms and intracellular stages [1]. The infection is initiated by sporozoite stages that are injected during mosquito feeding. The sporozoites invade the liver cells and undergo an asexual replication, called exoerythrocytic schizogony, that produces merozoites. Merozoites are released from the infected liver cells and invade erythrocytes. After invading the erythrocyte, the parasite initially develops into the ring stage followed by the trophozoite stage. In the case of *P. falciparum*, and other parasite species with 48-h replication cycles, the ring-stage lasts approximately 24 h. During the trophic period the parasite ingests host cell hemoglobin and produces hemozoin, also called the malarial pigment, as a waste product. Nuclear replication marks the beginning of the schizont stage and occurs approximately 36 h after merozoite invasion. During the schizont stage, there are multiple rounds of nuclear replication without cytoplasmic division. These newly replicated nuclei migrate to the parasite periphery and merozoites bud from the schizont. The infected erythrocyte is lysed following merozoite formation, and the released merozoites invade erythrocytes and repeat this blood-stage replication cycle. These repeated rounds of asexual replication in erythrocytes maintain the infection. Instead of repeating this asexual replicative cycle, some merozoites develop into sexual forms called gametocytes. Gametocytes are infective to mosquitoes and initiate sporogony in the mosquito. The sporozoites produced during sporogony migrate to the salivary glands of the mosquito and complete the life cycle. 

Disease manifestations associated with malaria are caused by the blood-stage replicative cycle, whereas the liver and gametocyte stages do not cause any significant pathology. The primary manifestations of malaria are periodic febrile paroxysms caused by the synchronous release of merozoites from infected erythrocytes. If untreated, these paroxysms generally last for a few weeks or months until immunity develops. Falciparum malaria exhibits higher morbidity and mortality than malaria caused by the other human *Plasmodium* species. This increased virulence associated with *P. falciparum* is also associated with higher parasitemia. One factor involved in the higher parasitemia of *P. falciparum* infection is the parasite’s ability to invade all erythrocytes, whereas *P. vivax* and *P. ovale* prefer immature erythrocytes and *P. malariae* prefer senescent erythrocytes [2]. *P. falciparum* also has a greater replication potential since mature schizonts produce more merozoites than the schizonts of the other human malaria parasites. 

Another important factor associated with *P. falciparum* virulence is the sequestration of infected erythrocytes in the deep tissues. This sequestration of infected erythrocytes is due to the cytoadherence of infected erythrocytes to endothelial cells of the capillaries, thereby decreasing blood perfusion, and causing localized inflammation. Cytoadherence is mediated by electron-dense protuberances on the surface of infected erythrocytes called knobs (Figure 1). These knobs were described in the early days of ultrastructural research on the malaria parasite [3], and the importance of knobs in the pathophysiology of severe falciparum malaria has been known for decades [4,5]. 

Knobs are supramolecular structures composed of parasite proteins exported to the host erythrocyte membrane and associated with a rearrangement of the erythrocyte’s submembrane cytoskeleton [6]. These knobs are the focal points for parasite ligands expressed on the erythrocyte surface and these ligands mediate the cytoadherence of infected erythrocytes to endothelial cells. Thus, the infected erythrocytes sequester in deep tissues and do not circulate through the spleen. This spleen avoidance is one means of immune evasion exhibited by the parasite [7]. In addition to immune evasion, sequestration also provides a better environment for parasite replication. For example, the release of merozoites in the capillaries increases invasion efficiency, and the low oxygen concentration in the deep tissues is a metabolically favored environment for the micro-aerobic parasite. Sequestration also contributes to the pathology and increased virulence associated with falciparum malaria [8].

Understanding the biology of knobs, cytoadherence, and sequestration-mediated pathology may provide insights into the management of severe falciparum malaria. This paper describes the structure and formation of knobs at the molecular and cellular levels and the nature of receptor-ligand interactions involved in cytoadherence, as well as the pathophysiology associated with cytoadherence.

## 2. Knob Structure

Knobs are electron-dense protuberances on the surface of *P. falciparum*-infected erythrocytes (see Figure 1). The electron-dense material making up the knob forms a cup-like structure 40–145 nm in diameter and approximately 50 nm in height [9]. It was recognized quite early that a parasite protein was correlated with knobs [10]. Sequencing of the gene for this knob-associated protein revealed three domains of repetitive sequence that are rich in lysine or histidine [11]. This knob-associated histidine-rich protein (KAHRP) is a major component of the knobs and is localized to the cytoplasmic face of the erythrocyte membrane (Figure 2). In addition, a spiral filament of unknown protein composition is also observed within the cup of the knob and forms a scaffold that binds KAHRP [9,12]. It is estimated that 60 KAHRP molecules make up a single knob [13] and KAHRP is likely the electron dense material of the knob. 

### 2.1. KAHRP Binds to the Submembrane Cytoskeleton of the Host Erythrocyte and May Mediate Its Reorganization 

On the cytoplasmic face of the erythrocyte membrane is a meshwork of filamentous proteins that forms a submembrane cytoskeleton. This cytoskeleton is responsible for the erythrocyte shape and for the deformability exhibited by erythrocytes. The meshwork is largely composed of filamentous proteins called spectrin and short actin filaments that are attached to integral erythrocyte membrane proteins via other proteins [14]. KAHRP binds to spectrin and other proteins of the submembrane cytoskeleton [6]. Binding to the spectrin skeleton is mediated by one of the histidine-lysine rich repeat regions of KAHRP since deletion of this repeat by genetic engineering abrogates normal knob formation [15]. This histidine-lysine repeat region is positively charged and binding to negatively charged spectrin likely involves electrostatic interactions [16]. 

Interaction of KAHRP with the submembrane cytoskeleton promotes a reorganization of the spectrin skeleton [9,12,13]. For example, the short actin filaments of the submembrane cytoskeleton are reorganized into long actin filaments that connect to the knobs [17]. Spectrin filaments connect to the knobs without passing through or under the knobs [9]. KAHRP appears to play an important role in cross-linking the knobs to the submembrane cytoskeleton. When the knobs reach high density, they are spaced in regular intervals that coincide with the extended length of the spectrin tetramer and the lattice formed by these filaments [15,18]. This rearrangement of the submembrane cytoskeleton also stiffens the spectrin skeleton and affects the rheological properties of infected erythrocytes [9,19]. For example, infected erythrocytes are less deformable than uninfected erythrocytes. 

### 2.2. The Major Cytoadherence Ligand Is Anchored into the Knobs

Another major parasite protein associated with knobs is *P. falciparum* erythrocyte membrane protein-1 (PfEMP1), the primary ligand associated with cytoadherence and severe disease [20]. PfEMP1 is localized to the knobs and is not distributed over the entire erythrocyte surface. The protein consists of a large variable N-terminal domain that is expressed on the erythrocyte surface, a transmembrane domain that transverses the erythrocyte membrane, and a conserved C-terminal domain that is localized to the cytoplasmic face of the erythrocyte membrane (Figure 3). This intraerythrocytic conserved domain has a high proportion of glutamic acid and aspartic acid residues and is called the acidic terminal segment (ATS). 

The ATS of PfEMP1 binds to both KAHRP and the spectrin lattice. The positively charged lysine and histidine rich repeat regions of KAHRP interact with the negatively charged ATS of PfEMP1 [21] via electrostatic interactions [22]. These electrostatic attractive forces may promote the complex formation leading to an accumulation of PfEMP1 at the knobs [23]. The ATS of PfEMP1 also interacts with submembrane cytoskeleton [16,24]. Thus, both PfEMP1 and KAHRP bind to components of the submembrane cytoskeleton, and these interactions may provide more stability by firmly anchoring PfEMP1 into the knobs. In addition, these interactions may cluster PfEMP1 at the apex of the knobs and promote cytoadherence by raising the receptor-binding domains of PfEMP1 above the glycocalyx of the erythrocyte. 

### 2.3. Other Parasite Proteins Are Associated with Knobs

There are also other parasite proteins that may be associated with knobs that have not been extensively characterized. These include knob associated heat shock protein 40 (HSP40), PfEMP3 [25], and a couple of proteins in the *Plasmodium* helical interspersed sub-telomeric (PHIST) family. PHIST member PFE1605w, also known as lysine-rich membrane-associated PHISTb, binds to the ATS of PfEMP1 and localizes to knobs [26,27]. The ATS also associates with PFI1780w, another member of the PHIST family [28]. Both PFI1780w and PFE1605w localize to the infected erythrocyte membrane, but only PFE1605w appears to localize to knobs [26]. PFE1605w may also interact with host erythrocyte membrane proteins [29]. It is also suggested that individual PHIST proteins may be optimized for individual PfEMP1 variants within the PfEMP1 gene family [29]. Proteomic studies of the knobs have identified other possible proteins of the knobs that have not yet been further investigated [30]. 

## 3. Knob Formation 

Starting at approximately 16 h post-merozoite invasion, knobs begin to appear on the surface of infected erythrocytes and increase in density as the parasite matures [31]. The expression of PfEMP1 on the erythrocyte surface is highly dependent on knobs since the amount of PfEMP1 on the erythrocyte surface is substantially decreased in knobless parasites [32]. Knob formation is a complex process in which KAHRP plus other parasite proteins are transported to the host erythrocyte membrane and assembled into knobs. KAHRP appears as punctate structures associated with the spectrin-actin meshwork at 16 h post-merozoite invasion [12]. These punctate structures enlarge and develop into knobs. Similarly, PfEMP1 also starts appearing on the erythrocyte surface during this same time frame [33]. Therefore, sequestration of infected erythrocytes does not begin until the parasites have progressed approximately halfway through the 48-h replicative cycle, coinciding with the period in which the ring stages are becoming trophozoites. It has long been recognized that only ring-stage infected erythrocytes are found in the peripheral circulation of individuals infected with *P. falciparum*, and erythrocytes infected with trophozoite and schizont stages are sequestered in the tissues. 

### 3.1. Unique Protein Trafficking Mechanisms Are Used by the Malaria Parasite to Modify the Host Erythrocyte 

Although the timing of the expression of knobs and PfEMP1 on the erythrocyte membrane are well known, the mechanisms by which the parasite alters the erythrocyte are not completely understood. The malaria parasite exports numerous proteins into the host erythrocyte to better accommodate the parasite and facilitate its survival and replication. These proteins are exported into the host erythrocyte via trafficking pathways with unique features [34,35,36]. For example, some exported parasite proteins have a unique targeting sequence called the *Plasmodium* export element (PEXEL), which is found in other apicomplexans and some stramenopiles [37,38]. Both KAHRP and PfEMP1 have PEXEL signals [39]. However, there are many exported parasite proteins that do not have PEXEL sequences [40,41]. Thus, there are multiple mechanisms by which parasite proteins are exported into the host erythrocyte. 

The host erythrocyte is devoid of organelles and the endomembranes of the endocytic and secretory pathways. Therefore, the malaria parasite cannot utilize the host pathways to modify the host cell, as is the case for many viruses and other pathogens. This raises questions about how the malarial parasite targets proteins to specific locations within the infected erythrocyte. Various membranous tubules and whorls originating from the parasitophorous vacuolar membrane (PVM) are observed in the cytoplasm of infected erythrocytes and these structures may be involved in the trafficking of exported parasite proteins within the host erythrocyte [38,42]. However, the variable expression of these membranous structures in the cytoplasm of infected erythrocytes, sometimes called the tubovesicular network (TVN), raises questions about their role in protein trafficking. 

Another membranous structure found in the host cytoplasm of *P. falciparum*-infected erythrocytes are the Maurer’s clefts [43]. The Maurer’s clefts are derived from the PVM and exhibit a flattened disk shape that is electron dense (see Figure 1). In addition, several parasite proteins that may play a role in the formation of the Maurer’s clefts or that may be residents of the Maurer’s clefts have been described (Table 1). Initially the Maurer’s clefts are mobile, and as the parasite matures, the clefts migrate to the erythrocyte periphery and become tethered to the cytoplasmic face of the erythrocyte membrane [44,45]. It is widely believed that Maurer’s clefts play a role in the movement of parasite proteins, especially PfEMP1, through the host erythrocyte cytoplasm to the erythrocyte membrane [46].

### 3.2. KAHRP Binds to the Submembrane Cytoskeleton of the Erythrocyte as a First Step in Knob Formation 

The submembrane cytoskeleton consists of a two-dimensional lattice of spectrin filaments on the cytoplasmic face of the erythrocyte membrane [14]. The unit structure of spectrin filaments are tetramers composed of two heterodimers of α- and β-spectrin, which are oriented in a head-to-head fashion. These spectrin tetramers are joined together at their tails to form the two-dimensional lattice via interactions with short actin filaments and a protein called band 4.1R (Figure 4). This 4.1R–actin complex attaches to glycophorins, which are integral membrane proteins that span the lipid bilayer. A protein called ankyrin binds to the spectrin tetramer near the head-to-head junction, and ankyrin attaches the spectrin skeleton to the anion exchanger called band 3. The complex between ankyrin, band 3, and spectrin is often called the ankyrin complex. This structural organization of the erythrocyte membrane allows the erythrocyte to undergo large deformations as they pass through small spaces, while maintaining the integrity of the erythrocyte. 

KAHRP does not appear to traffic via the Maurer’s clefts en route to the erythrocyte membrane [48]. Presumably, KAHRP is secreted as a soluble protein that directly binds to the submembrane cytoskeleton where it may nucleate the formation of knobs. For example, KAHRP binds to both spectrin [54,55] and ankyrin [54,56,57]. Binding of KAHRP to ankyrin is near the band 3-binding domain of ankyrin, suggesting that KAHRP may interfere with the ankyrin complex [56]. Other possible KAHRP binding sites for knob nucleation include the 4.1R–junctional complex [24], the ankyrin bridge between spectrin and the band 3 anion transporter [16], or the meshwork formed by spectrin filaments [12]. It is not clear if all three of these KAHRP-binding sites are involved in the initiation of knob formation. It has been suggested, though, that eventually KAHRP co-localizes with actin filaments making up the band 4.1R–junctional complex and this is the nexus of knob formation [13].

### 3.3. Chaperones and Other Parasite Proteins May Assist in the Movement of PfEMP1 to the Host Erythrocyte Membrane

At least eight genes have been identified that are involved in the export of PfEMP1 to the erythrocyte membrane and the formation of knobs [58]. These proteins have distinct roles in the various steps involved in the movement of PfEMP1 from the parasite to the erythrocyte membrane (see Table 1). Many of these proteins are chaperones and several parasite chaperones are exported into the host erythrocyte [59,60]. Chaperones promote the efficient folding of proteins, prevent protein aggregation, can assist in the intraerythrocytic trafficking of proteins, or participate in the translocation of proteins across membranes. Many chaperones are designated as heat-shock proteins (HSPs) and include the HSP40, HSP70, and HSP90 families. The malaria parasite has several members of each of these families [61]. PfHSP70-x is a HSP70 family member that is exported into the host erythrocyte [62] and several members of the HSP40 family, also called J-domain proteins, are exported into the host erythrocyte [63]. 

PfEMP1 is likely secreted from the parasite as a soluble protein that initially interacts with chaperone-containing complexes called J-dots [49]. PfHSP70-x and an exported parasite J-domain protein are likely involved in the formation of J-dots. These J-dots are mobile molecular complexes found within the cytoplasm of infected erythrocytes, and the J-dots presumably carry PfEMP1. However, J-dots do not transport PfEMP1 to the erythrocyte membrane and PfEMP1 is likely transferred from J-dots to Maurer’s clefts [64,65]. Maurer’s clefts with their PfEMP1 cargo become tethered to the erythrocyte membrane, and with the assistance of other parasite proteins, PfEMP1 is translocated from Maurer’s clefts to the knobs. 

### 3.4. Knobs Are Assembled In Situ at the Erythrocyte Membrane

The direct binding of KAHRP to the submembrane cytoskeleton suggests that the knobs are assembled in situ at the erythrocyte membrane and the binding of the KAHRP to the submembrane cytoskeleton is the first step in knob formation. KAHRP is also known to self-associate into large complexes [24] and this self-association could promote an accumulation of KAHRP at the 4.1R–junctional complex. There is also a rearrangement of the submembrane cytoskeleton that occurs during knob formation, but no specific information is available as to the mechanism of this rearrangement. In addition, a yet unknown protein is then possibly recruited and assembles the spiral complex making up the knob scaffold [9]. 

PfEMP1 is also incorporated into knobs after they have been at least partially formed [12]. The delivery of PfEMP1 from the Maurer’s clefts to the erythrocyte membrane may involve the insertion of PfEMP1 into cholesterol-rich microdomains, called lipid rafts [66]. Specific parasite proteins (see Table 1), such as PfEMP3 [51] or PfEMP1 trafficking protein-7 [52], have also been implicated in the translocation of PfEMP1 from the Maurer’s clefts to the knobs. Although the exact mechanism of translocation of PfEMP1 across the erythrocyte membrane is not known, the process does appear to be inefficient since less than 10% of the total PfEMP1 within the infected erythrocyte is exposed on cell surface [51]. The ATS of PfEMP1 also interacts with the 4.1R–junctional complex of the erythrocyte submembrane cytoskeleton [16,24]. This interaction with both KAHRP and the 4.1R–junctional complex may also help to assemble PfEMP1 into the knob.

## 4. Var Genes and PfEMP1 Variants 

High molecular weight proteins involved in strain-dependent adherence to endothelial cells were first identified in the 1980s [67,68] and subsequently named PfEMP1 [69]. Cloning and sequencing studies revealed that PfEMP1 is a member of a variable gene family called var [70]. The identification of the var gene family helped clarify the apparent strain specificity of cytoadherence and the numerous potential endothelial cell receptors involved in cytoadherence (discussed below). Members of the PfEMP1 family range in size from 200 to 450 kDa and are anchored into the knobs on the surface of *P. falciparum*-infected erythrocytes (see Figure 2). There are 47–90 PfEMP1 genes per parasite genome [71] and most parasite isolates have approximately 60 PfEMP1 alleles. Individual parasite lineages have different repertoires of PfEMP1 genes due to frequent recombination between PfEMP1 alleles [72]. PfEMP1 genes are found in sub-telomeric regions of all chromosomes and in the interior regions of some chromosomes. Other variant surface antigens encoded by multigene families include sub-telomeric variable open reading frame (STEVOR) proteins, and repetitive interspersed family (RIFIN) proteins [73]. By far, PfEMP1 is the most important among these variant surface antigens in regard to cytoadherence and severe disease [20]. The gene families of proteins expressed on the erythrocyte surface provide a means for the parasite to evade the immune system by undergoing antigenic variation. 

### 4.1. Subtypes of DBL and CIDR Modules Define the Variation between PfEMP1 Alleles 

PfEMP1 genes consist of two exons (see Figure 3). The first exon encodes the highly variable extracellular domain of PfEMP1, and the second exon encodes a short trans-membrane domain plus a conserved C-terminus of the protein called the ATS. The ATS interacts with KAHRP and the submembrane cytoskeleton on the cytoplasmic side of the erythrocyte membrane (discussed above). The extracellular variable domain exhibits a modular structure consisting of 2–10 Duffy-binding-like (DBL) domains and 1–2 cysteine-rich interdomain regions (CIDRs). The DBL domains exhibit sequence homology to adhesive domains in *P. falciparum* erythrocyte-binding antigen (EBA)-175 and the Duffy-binding proteins of *P. vivax* and *P. knowlesi* [74]. EBA-175 and Duffy-binding proteins are adhesion proteins that mediate the attachment of merozoites to receptors on the erythrocyte surface as an early step in host cell invasion [75]. 

Based on sequence phylogeny, the various DBL domains can be divided into seven distinct types designated as α, β, γ, δ, ε, ξ, or χ. CIDR domains are characterized by conserved cysteine-rich motifs [76] and exhibit three distinct sequence types designated as α, β, or γ. These various DBL and CIDR types are further divided into subtypes that are designated with numbers following the Greek letters [77]. Each PfEMP1 allele is characterized by a unique combination and order of DBL and CIDR sequence types. 

### 4.2. Specific Arrangements of DBL and CIDR Modules Define PfEMP1 Domain Cassettes

The combination of DBL and CIDR domains in any given PfEMP1 protein is not completely random, but certain modules tend to occur together. For example, following an N-terminal sequence, more than 75% of the PfEMP1 alleles have a DBLα domain immediately followed by a CIDRα domain [77,78]. This combined DBLα/CIDRα head group is followed by the other DBL and CIDR sequence types. Certain combinations of subtypes of the DBL and CIDR domains tend to be found together, forming domain cassettes [77]. For example, the common head group consisting of DBLα2 and CIDRα1 followed by DBLβ12 and DBLγ4/6 makes up domain cassette (DC) 8. Thus, domain cassettes are analogous to haplotypes. Three unique var genes that are not part of a domain cassette have also been identified. These have been named var1, var2csa, and var3 and all three contain unique DBL and CIDR modules that are not found in other PfEMP1 alleles [77]. In addition, var2csa and var3 lack the common DBLα/CIDRα head group that is found at the beginning of most other PfEMP1 alleles. Orthologs of var1 and var2csa genes are found in *P. reichenowi* [79], a malaria parasite of chimpanzees that is closely related to *P. falciparum*. 

## 5. Antigenic Variation 

The expression of PfEMP1 exhibits an antigenic variation in which different PfEMP1 alleles are sequentially expressed. PfEMP1 alleles are antigenically distinct, and this sequential expression allows *P. falciparum* to avoid antibody-mediated destruction of infected erythrocytes since the antibodies against one PfEMP1 allele do not recognize other PfEMP1 alleles. The intrinsic switch rate in var gene expression may occur approximately every 50 blood-stage generations [80]. Therefore, antigen switching is a form of immune evasion that hampers the development of an effective immune response against PfEMP1. The resulting inability to eliminate infected erythrocytes maintains long-term chronic infections characterized by waves of parasitemia. It has long been recognized that immunity against malaria is slow to develop and requires multiple exposures [20,81]. Part of this slow development of immunity is due to antigenic variation and the need to develop antibodies against multiple PfEMP1 alleles. 

### 5.1. Epigenetic Mechanisms and cis-DNA Elements Are Involved in the Monoallelic Expression of PfEMP1 Alleles

Only a single *var* gene is expressed by an individual parasite and this monoallelic expression, also called mutually exclusive expression, is a complex process that is not completely understood [82,83]. By default, the expression of var genes is repressed and only a single allele is expressed during the blood stage replication cycle [84]. There are several levels of regulation of PfEMP1 expression that include: (1) histone modifications, (2) promoter/intron interactions (i.e., cis-DNA element), (3) non-coding RNA transcripts, and (4) nuclear architecture (Figure 5). The expressed PfEMP1 allele, along with the appropriately modified histones, cis-DNA elements, and non-coding transcripts, is localized to a specific expression site on the nuclear membrane, whereas non-expressed alleles are found at other nuclear membrane locations. The activation of PfEMP1 involves the movement of a PfEMP1 allele from a non-expression site to the expression site [85,86]. 

Chromatin can exist in two states called euchromatin and heterochromation. Heterochromatin is highly condensed and transcriptionally inactive, whereas euchromatin is more open and capable of being transcribed [87]. The transition between euchromatin and heterochromatin is regulated in part by the post-translational modification of proteins, called histones, that directly interact with DNA. These histone modifications primarily involve methylations and acetylations, and are often called the histone code [88]. Histone methylations or acetylations that are associated with either gene repression or gene activation have been described in conjunction with the expression of PfEMP1 [81,82]. These modifications can be retained after cell replication and the status of the chromatin is maintained through the generations. This inheritance of chromatin patterns is called epigenetics and epigenetics maintain the expression of a specific PfEMP1 allele for several blood-stage replicative cycles until the expressed allele is switched. The precise factors that control this switching have not yet been described. 

### 5.2. Transcripts Originating from PfEMP1 Introns Regulate Gene Expression

The introns of *var* genes are highly conserved between alleles and the intron regulates the expression of PfEMP1 on two levels. By mechanisms that are not completely understood, there is a pairing of a PfEMP1 intron and its promoter at the nuclear expression site that results in transcription silencing [89]. In addition, the intron has a bi-directional promoter that controls the transcription of long non-coding RNA (lncRNA) transcripts [90]. The sense lncRNA is associated with the repression of transcription, whereas the lncRNA corresponding to the antisense strand is associated with the activation of expression [91,92]. However, no specific transcription factor that activates the expression of *var* genes has been identified. There is still a lot to be learned about the mechanisms that maintain monoallelic expression, the activation of expression from the repressed state, and the coordination to ensure that the activation of one allele coincides with the silencing of the previously active allele.

## 6. Cytoadherence Receptors

Proteins or glycosaminoglycans expressed on the surface of endothelial or other cells serve as cytoadherence receptors, and these receptors are responsible for the sequestration of *P. falciparum*-infected erythrocytes in the capillary beds of various organs [93]. Numerous potential receptors have been implicated in the cytoadherence of infected erythrocytes to endothelial cells (Table 2). These potential endothelial cell receptors (ECRs) have primarily been identified by in vitro assays that measure the binding of infected erythrocytes to cells expressing the candidate ECR or binding of infected erythrocytes to purified receptors. After the discovery of PfEMP1, the direct binding of the receptor with PfEMP1 has been confirmed for some of the proposed ECR receptors, but not all.

Many of these potential receptors were identified before the description of PfEMP1 as the major cytoadherence ligand and the discovery of the *var* gene family. Initially, the large number of potential receptors and the strain specificity of receptor binding were a bit of an enigma since receptor–ligand interactions are generally viewed as highly specific. The realization that PfEMP1 was part of the multi-gene family helped clarify this enigma. The various PfEMP1 alleles have different binding phenotypes in that the different PfEMP1 alleles recognize different ECR. Therefore, as the parasite undergoes antigenic variation, there is a corresponding change in the specific receptor that is recognized. This means that each PfEMP1 allele potentially has a unique cytoadherence phenotype. 

### 6.1. Endothelial Cell Receptors Include Adhesion Molecules and Proteins Associated with Inflammation

Many of the potential ECRs are adhesion molecules that participate in cell–cell interactions, especially interactions between leukocytes and endothelial cells, which occur during inflammation. For example, ECRs include several members of the immunoglobulin-like superfamily, selectins, integrin, complement receptor 1, gC1qR, and fractalkine. These proteins participate in adhesive phenomenon and are often associated with inflammation (see Table 2). In addition, some of these potential ECRs, such as endothelial protein C receptor (EPCR), platelet endothelial cell adhesion molecule 1, E-selectin, P-selectin, and thrombospondin, are also associated with platelet activation and aggregation, as well as inflammation. 

The better characterized ECRs are CD36, EPCR, and intercellular adhesion molecule-1 (ICAM1). CD36 was one of the first described ECRs and has been extensively characterized [122]. This receptor has also been widely used in the study of cytoadherence. CD36 is a glycoprotein that is expressed on the surface of many cell types and on endothelial cells of liver, spleen, skin, lung, muscle, and adipose tissue [96]. EPCR is a protein on the surface of endothelial cells that binds to protein C and promotes the protein C anti-coagulant pathway, and thereby regulates thrombosis and inflammation [98]. ICAM-1 is a transmembrane glycoprotein that belongs to the immunoglobulin-like superfamily. In addition to ICAM1 [99], other members of the immunoglobulin-like superfamily that are potential ECR include vascular cell adhesion molecule-1 [101], platelet endothelial cell adhesion molecule-1 [78], and neural cell adhesion molecule [104]. ICAM1 is constitutively expressed on a wide range of cell types, including endothelial cells, and is upregulated in response to proinflammatory cytokines such as tumor necrosis factor-α (TNF-α) [100].

### 6.2. DBL and CIDR Domains Determine Cytoadherence Phenotype

The various alleles of PfEMP1 exhibit different phenotypes in regard to the endothelial cell receptors that they recognize. This cytoadherence phenotype is largely due to the subtypes of DBL and CIDR adhesion domains that make up a specific PfEMP1 allele [8,20,122]. In other words, receptor binding is often associated with a single adhesive domain, or a combination of adhesive domains found in domain cassettes (Table 3). For example, binding to CD36 is due to the common DBLα/CIDRα head group [78,123]. This explains the large number of PfEMP1 variants that bind CD36 since more than 75% of PfEMP1 variants contain this common head group. 

Similarly, a CIDRα1 module in conjunction with DC8 [97] or DC13 [134] mediates binding to EPCR and cytoadherence of infected erythrocytes to endothelial cells. Likewise, ICAM-1 binding is mapped to the first DBLβ domain that follows the PfEMP1 common head group in DC4 and DC8 [134]. In members of DC8 and DC13, each DBLβ domain that is predicted to bind to ICAM-1 is also preceded by an EPCR-binding CIDRα1 domain in the common head group. This means that the alleles of PfEMP1 that are members of DC8 or DC13 bind to both EPCR and ICAM1 [126,134]. This implies a synergy between EPCR and ICAM1 binding. A synergy between CD36 binding and ICAM1 binding has also been suggested [135,136].

### 6.3. A Single PfEMP1 Allele Binds to Chondroitin Sulfate A in the Placenta

Another well characterized cytoadherence receptor is chondroitin sulfate A (CSA), which is a glycosaminoglycan found in the placenta [117,118]. Glycosaminoglycans are linear polysaccharides consisting of repeating disaccharide units that are expressed on cell surfaces and the extracellular matrix [137]. Other glycosaminoglycans that have been implicated as cytoadherence receptors include hyaluronic acid [119] and heparan sulfate [120]. CSA and hyaluronic acid are secreted by the placental cells that line the intervillous spaces, making up an area of extensive contact between fetal tissue and maternal circulation. These glycosaminoglycans may help protect against host-vs-graft rejection by masking fetal antigens from the maternal immune system. 

A single PfEMP1 allele, called var2csa, is the ligand that binds CSA [138,139] and hyaluronic acid [140]. Var2csa is a unique PfEMP1 allele that does not belong to a domain cassette group and exhibits unique DBL and CIDR domains [77]. The modular structure of var2csa is DBLu–DBLu–CIDRu–DBLu–DBLε4–DBLε4–DBLε10, where the lowercase u signifies unique subtypes of DBL and CIDR modules. Binding to CSA has been mapped to the second DBLu domain [133]. Parasites expressing var2csa on the erythrocyte surface sequester in the placenta and are likely responsible for placental malaria. 

### 6.4. PfEMP1 Mediates Other Cytoadherence Phenomena

Two other phenomena that are linked to PfEMP1-mediated cytoadherence are rosetting and platelet-mediated clumping. Rosetting refers to the adherence of uninfected erythrocytes to infected erythrocytes to form a relatively large clump of cells with an infected erythrocyte in the center that is surrounded by uninfected erythrocytes [141]. PfEMP1 is involved in some rosetting, but not all, since other erythrocyte surface antigens also mediate rosetting. Three potential PfEMP1 receptors associated with rosetting have been identified: complement receptor 1 [112], heparin sulfate [120], and type A or type B blood group antigens [121]. DBLα domains have been implicated in receptor–ligand interactions for all three of these receptors (see Table 3). 

Platelets have also been implicated in the formation of erythrocyte clumps during *P. falciparum* infection [142]. Infected erythrocytes can bind to platelets and the platelets function as bridges between infected erythrocytes [143]. Platelets also bind to endothelial cells and can function as bridges between infected erythrocytes and endothelial cells [144]. Three potential PfEMP1 receptors for platelet mediated clumping are CD36 [143,144], gC1qR/HABP1/p32 [114], and P-selectin [145]. Other potential receptors that are associated with platelets include platelet endothelial cell adhesion molecule-1, E-selectin, and thrombospondin. Blood coagulation dysfunction, and especially activation of coagulation, is often observed during severe malaria [146]. Disseminated intravascular coagulation is observed in approximately 15% of severe falciparum malaria cases and in approximately 80% of the fatal malaria cases [147]. 

## 7. Severe Falciparum Malaria

Clinical manifestations associated with malaria can range from asymptomatic carriage to a febrile illness to severe organ dysfunction resulting in death [148]. Febrile illness without organ dysfunction is often called uncomplicated malaria and exhibits non-specific symptoms such as fever, nausea, and headache. Complicated malaria refers to severe malaria associated with organ dysfunction. Approximately one percent of falciparum malaria cases develop into severe disease [148,149]. Case fatality rates of severe falciparum malaria range from 5% to 50%, depending on the extent of organ involvement and availability of therapy. Multiple organs can be affected during complicated malaria and the manifestations are determined by the affected organ(s) (Table 4). Patients can exhibit several of these manifestations either simultaneously or sequentially. In addition, children and adults often exhibit different severe manifestations of malaria [150]. The prognosis of individuals exhibiting any of the indicators of complicated malaria is poor, and without treatment the risk of death increases.

Several factors are involved in the development of severe malaria [151]. These include host genetics, patient age, and prior exposures to *P. falciparum*, which all can affect parasitemia. It is widely accepted that hyperparasitemia is associated with a poor prognosis and the total parasite biomass correlates with severe disease [152]. However, some children can tolerate extremely high parasitemia without exhibiting severe disease [153]. Generally, exposure-dependent acquisition of immunity decreases the development of severe malaria, but there are exceptions [154]. There is still much to be learned about severe falciparum malaria and predicting the clinical course of malaria is difficult. 

### 7.1. Cerebral Malaria, Respiratory Distress, and Severe Anemia Are Common Manifestations of Complicated Malaria

The three most common manifestations of severe falciparum malaria are cerebral malaria, respiratory distress, and severe anemia in children [155]. Cerebral malaria is characterized by an impaired consciousness and other neurological symptoms [156]. Patients typically present with fever and severe headache for several days followed by drowsiness, confusion, repeated seizures, convulsions, and ultimately an unrousable coma. Swelling of the brain and hemorrhage into the brain tissue are also present. Common co-morbidities include respiratory distress, hypoglycemia, and acidosis. Metabolic acidosis, as manifested by respiratory distress, has emerged as a central feature of severe falciparum malaria and is a better predictor of death than cerebral malaria or severe anemia [148,157]. The first signs of lung injury are rapid and difficult breathing accompanied by pulmonary edema. This pulmonary edema can progress to acute respiratory distress syndrome and even respiratory failure. 

Severe anemia is due to both the increased destruction of erythrocytes and decreased production of new erythrocytes [158]. In addition to the erythrocytes that are destroyed by the parasite during blood-stage schizogony as part of the parasite’s life cycle, non-infected erythrocytes are destroyed at higher rates due to complement-mediated lysis and phagocytosis mediated by immune complex deposition or complement activation. Furthermore, there is a decreased production of erythrocytes during infection, resulting in less replacement of the lost erythrocytes. Acute kidney injury is also associated with high mortality during severe falciparum malaria [159]. 

### 7.2. Sequestration of the Infected Erythrocytes in Microvasculature Is a Major Factor in Disease Pathogenesis 

The cytoadherence of parasitized erythrocytes to the capillary endothelium of vital organs, such as brain, bone marrow, lungs, kidneys, or intestines, certainly plays a major role in complicated falciparum malaria. The sequestration of parasitized erythrocytes results in microvascular obstruction, localized inflammation, increased vascular permeability, coagulation disorders, and ultimately organ dysfunction (Figure 6). 

The adherence of infected erythrocytes in the capillaries obstructs blood flow and deprives tissues of nutrients and oxygen. This ischemia may be enhanced by rosettes [160] or platelet-induced clumps [145]. In addition, the metabolism of the parasite further complicates ischemia. The parasite has a high demand for glucose and can lead to localized hypoglycemia. Furthermore, the parasite exhibits an anaerobic metabolism and converts glucose into lactic acid via glycolysis, thus promoting acidosis. Infection also increases the anaerobic metabolism of the host tissues, which further increases the hypoglycemia and acidosis [161]. Metabolic acidosis is a key feature of severe falciparum malaria and is a major contributing factor to respiratory distress. 

In addition to the obstruction of capillaries and ischemia, cytoadherence causes increased production of inflammatory cytokines—especially TNF-α—and endothelium inflammation is a major feature of severe malaria [20,162]. Activated endothelial cells also express higher levels of potential PfEMP1 receptors such as ICAM1, P-selectin, and E-selectin. This increased expression of cytoadherence receptors can enhance sequestration. The inflammatory cytokines also lead to endothelium dysfunction and increased vascular permeability in the affected tissues and organs. For example, pulmonary edema is a major feature of respiratory distress. Likewise, edema and the swelling of the brain due to the disruption of the blood–brain barrier is a common feature of cerebral malaria [163]. Inflammation also results in the activation of platelets and platelet accumulation is significantly higher in cerebral malaria patients than in uncomplicated malaria [164].

### 7.3. Expression of Specific PfEMP1 Alleles Is Associated with Severe Disease and Organ Specific Clinical Manifestations 

A logical inference from the wide range of endothelial cell receptors recognized by PfEMP1 is that different PfEMP1 alleles may be responsible for the organ specificity of the various complications associated with severe falciparum malaria [93,151]. Indeed, it has been demonstrated that parasites isolated from different organs express different variants of PfEMP1 [165]. For example, there are correlations between the expression of specific PfEMP1 alleles and the development of cerebral malaria or placental malaria. However, specific PfEMP1 alleles that target infected erythrocytes to the lungs, bone marrow, kidneys, or other organs associated with complicated malaria have not yet been identified. Nonetheless, it is likely that specific PfEMP1 alleles may have a tropism for a particular tissue or organ, and this organ specific tropism could account for the wide range of clinical manifestations that are associated with complicated malaria (see Table 4).

Several studies have shown that the expression of specific PfEMP1 alleles is correlated with the risk of developing severe disease [19,156,158]. For example, PfEMP1 alleles with EPCR binding capabilities tend to be associated with severe disease and alleles with CD36 binding capabilities tend to be associated with milder disease [122,166,167]. In particular, the expression of PfEMP1 of domain cassettes 8 or 13, which bind to both EPCR and ICAM1, are associated with cerebral malaria [168,169]. Consistent with a possible role in cerebral malaria, members of DC8 and DC13 exhibit preferential adherence to brain endothelial cells [170,171]. However, questions have been raised about the role of ICAM1-binding PfEMP1 alleles in cerebral malaria [172]. Heparan sulfate-binding alleles [173] and gC1qR-binding alleles [130,174] have also been implicated in severe disease. 

A good example of a specific PfEMP1 allele that is correlated with a specific disease manifestation is the association of var2csa expression with placental malaria. The expression of var2csa is significantly upregulated following the selection for adhesion to CSA in vitro [175]. Furthermore, var2csa is the predominant var gene that is expressed in parasites isolated from the placenta [133,176,177]. Placental sequestration impacts both mother and fetus, contributing to premature delivery, intrauterine growth retardation, stillbirth, maternal anemia, and increased neonatal and maternal mortality [178]. In addition, it is known that the severity of placental malaria decreases with subsequent pregnancies. This decrease in disease severity in multigravida women is likely due to antibodies directed against the var2csa variant of PfEMP1, which would develop as a result of acquiring malaria during a previous pregnancy [179]. Thus, multigravida women have better immunity directed at var2csa than primigravida women. In this regard, clinical trials with var2csa as a potential vaccine are underway [180]. 

## 8. Summary

PfEMP1 is a variable protein that is expressed on knobs at the surface of *P. falciparum*-infected erythrocytes, and PfEMP1 mediates cytoadherence. Cytoadherence to endothelial cells results in the sequestration of infected erythrocytes in capillary beds and diminishes the clearance of infected erythrocytes by the spleen. PfEMP1 is encoded by the var gene family, which includes approximately 60 alleles per genome. The expression of PfEMP1 also exhibits antigenic variation in which only a single PfEMP1 allele is expressed at a time. The mechanism of antigenic variation involves histone modifications, cis-DNA elements, non-coding transcripts, and nuclear membrane expression sites. Antigenic variation circumvents elimination of infected erythrocytes via antibody-mediated mechanisms directed at PfEMP1. 

PfEMP1 also functions as the cytoadherence ligand and binds to receptors expressed on the surface of endothelial cells. The various PfEMP1 alleles exhibit distinct cytoadherence phenotypes in regard to the specific endothelial cell receptors that are recognized. Therefore, a switch in the expressed PfEMP1 allele may be accompanied by a switch in the endothelial cell receptor that is recognized. The various endothelial cell receptors recognized by PfEMP1 alleles are differentially distributed in the various organs. This means that a change in the receptor-binding phenotype may also direct sequestration to different organs. A change in sequestration can influence clinical manifestations and the severity of malaria. Pathology associated with sequestration includes ischemia related phenomenon—such as localized hypoglycemia and acidosis—due to the obstruction of blood vessels, localized inflammation of the endothelium, and endothelium dysfunction leading to vascular leakage and edema. Decreased perfusion, acidosis, and vascular leakage all contribute to the higher mortality and morbidity associated with severe falciparum malaria.

A better understanding of the biology of cytoadherence and sequestration may lead to the development of therapeutic interventions. Drugs or other therapeutics that interfere with knob formation, expression of PfEMP1 on the erythrocyte surface, binding of PfEMP1 to ECR, or antigenic variation have the potential to reduce the severity of falciparum malaria. Much progress has been made in describing the molecular interactions involved in the knob formation and the receptor–ligand interactions mediating cytoadherence. Understanding the specifics of these interactions opens up the possibility of developing low molecular weight molecules that interfere with these processes. Detailed knowledge about the molecules involved in cytoadherence may also lead to the development of therapeutic vaccines. Furthermore, understanding how cytoadherence mediates the pathology associated with severe malaria can lead to better supportive care in patients with severe malaria. 

## Figures and Tables

**Figure 1 tropicalmed-08-00353-f001:**
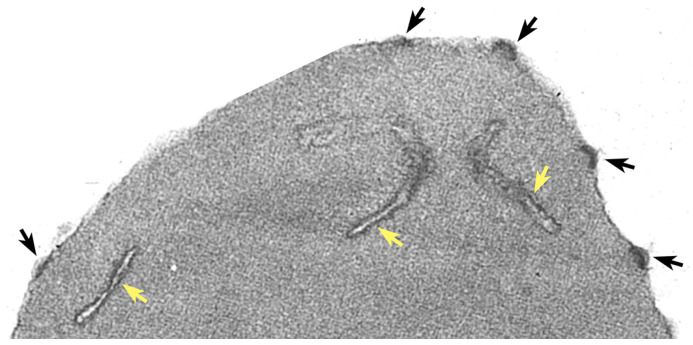
Ultrastructure of knobs and Maurer’s clefts. An electron micrograph of an infected erythrocyte showing knobs (black arrows) and Maurer’s clefts (yellow arrows). Electron micrograph provided by H. Norbert Lanners.

**Figure 2 tropicalmed-08-00353-f002:**
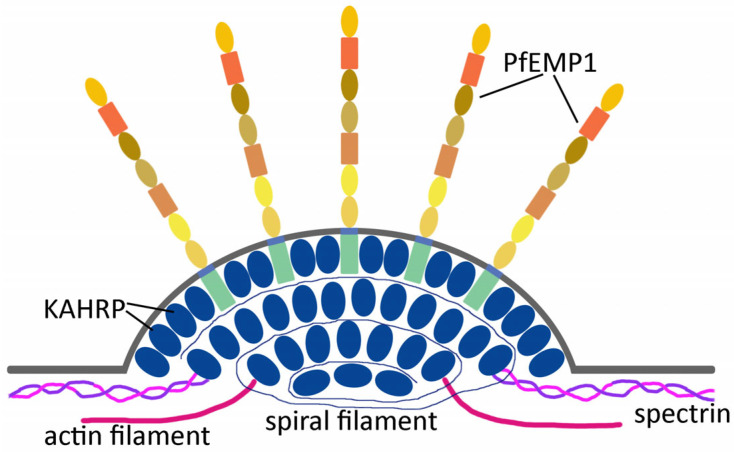
Schematic representation of knob components. A spiral filament of unknown composition provides the scaffold for knob formation and the association of knob-associated histidine rich protein (KAHRP). Spectrin and actin filaments are embedded in the knob on the cytoplasmic face of the erythrocyte membrane. PfEMP1, which functions as the cytoadherence ligand, is exposed on the erythrocyte surface. Interactions with KAHRP and the reorganized submembrane cytoskeleton of the erythrocyte anchors PfEMP1 into the knob.

**Figure 3 tropicalmed-08-00353-f003:**
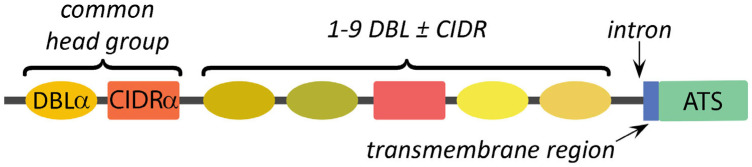
PfEMP1 structure. PfEMP1 consists of a large variable extracellular domain and a conserved intracellular domain (green), called the acidic terminal segment (ATS), that are separated by a transmembrane region (blue). The gene contains a single intron of approximately 1 kb that is located between the extracellular domain and the transmembrane region (arrow). A variable number of Duffy binding-like domains (DBLs) and cysteine-rich interdomain regions (CIDRs) make up the extracellular domain. Sequencing reveals distinct types of DBLs and CIDRs (portrayed with different colors). A common head group, defined by an alpha (α) type of DBL followed by an alpha (α) type of CIDR, is found in more than 75% of PfEMP1 alleles.

**Figure 4 tropicalmed-08-00353-f004:**
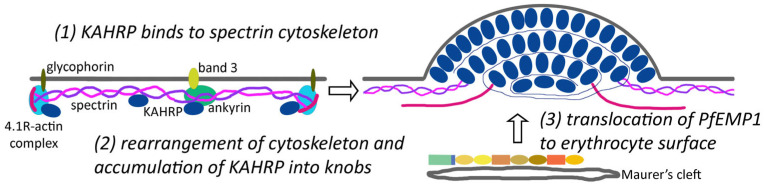
Steps in knob formation. Knob-associated histidine rich protein (KAHRP) is secreted from the parasite as a soluble protein and binds to the spectrin meshwork making up the submembrane cytoskeleton. Possible binding sites for KAHRP (dark blue ovals) include the 4.1R–actin complex (light blue ovals plus red line), the spectrin tetramer, and the ankyrin (green oval) complex. KAHRP promotes a reorganization of the submembrane cytoskeleton and KAHRP is recruited to form the knobs. (See Figure 2 for the annotation of the knob.) Coincident with the formation of the knobs, *P. falciparum* erythrocyte membrane protein-1 (PfEMP1) is translocated from the Maurer’s clefts to the knobs and exposed on the surface of the infected erythrocyte.

**Figure 5 tropicalmed-08-00353-f005:**
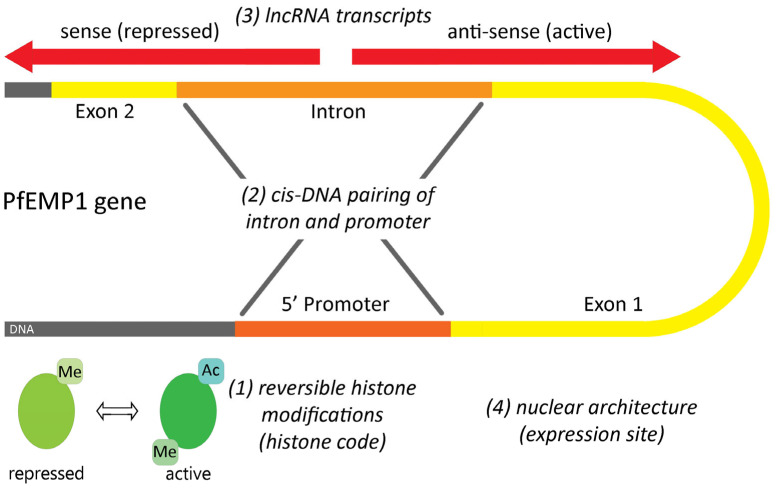
Potential factors regulating the transcription of PfEMP-1. Factors that regulate the expression of PfEMP1 include: (1) chromatin status as determined by histone methylations (Me) and acetylations (Ac), (2) cis-interactions between the intron and promoter of a PfEMP1 gene, (3) long non-coding transcripts (lncRNA) originating from the intron, and (4) location of the expressed PfEMP1 gene at a specific expression site on the nuclear membrane.

**Figure 6 tropicalmed-08-00353-f006:**
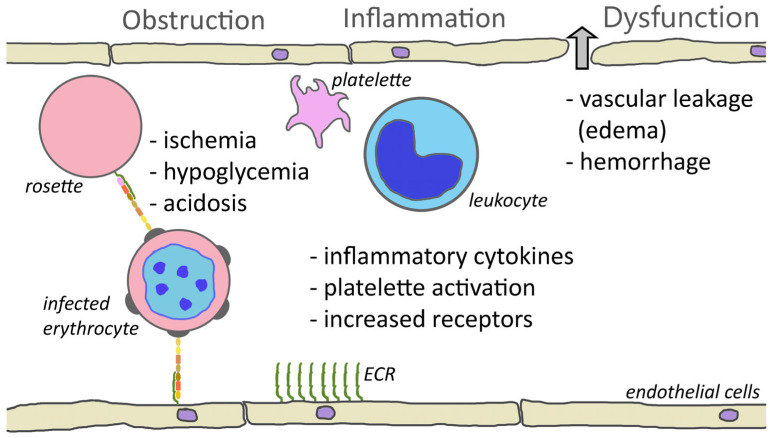
Cytoadherence-mediated pathogenesis. One consequence of cytoadherence is the obstruction of capillaries. Rosetting and platelet-mediated clumping may augment this obstruction. Obstruction of blood vessels results in reduced blood flow and ischemia. This decreased perfusion combined with parasite metabolism may contribute to hypoglycemia and acidosis. Cytoadherence also inflames the endothelium and attracts leukocytes and activates platelets. This inflammation and increased levels of inflammatory cytokines cause a loosening of the tight junctions between endothelial cells and results in leakage of fluid into the tissues or hemorrhaging (block arrow). Inflammation also increases the expression of endothelial cell receptors (ECRs) and may augment cytoadherence.

**Table 1 tropicalmed-08-00353-t001:** Parasite proteins possibly involved in knob assembly and the transport of parasite proteins to the knobs.

Protein	Possible Role
PfSBP1 [47]	May function in the formation of the Maurer’s clefts at the parasitophorous vacuole membrane and the loading of PfEMP1 onto the clefts.
MAHRP1 [48]	Plays a role in the export of PfEMP1 from the parasite and its association with the Maurer’s clefts.
PfHSP70-x [49]	A HSP70 unique to *P. falciparum* that is exported to the host erythrocyte where it forms a complex with PfHSP40 and PfEMP1 in structures known as J-dots.
PfPTP1 [50]	Disruption of this gene leads to morphological abnormalities of the Maurer’s clefts and prevents the expression of PfEMP1 on the erythrocyte surface.
PfEMP3 [51]	May function in the translocation of PfEMP1 from the Maurer’s clefts to the surface of the erythrocyte.
PTP7 [52]	Required for recruitment or formation of vesicles at the Maurer’s clefts and the transfer of PfEMP1 to the host erythrocyte membrane.
PFE1605w [26]	A member of the PHIST family that binds the ATS of PfEMP1 and possibly comigrates with PfEMP1 during trafficking within the infected erythrocyte.
PFA66 [53]	A J-domain protein (i.e., HSP40) that is needed for the proper assembly of knobs and the expression of PfEMP1 on the erythrocyte surface. May also interact with host erythrocyte HSP70.

PfEMP = *P. falciparum* erythrocyte membrane protein; PfSBP1 = *P. falciparum* skeleton-binding protein-1; MAHRP1 = membrane-associated histidine-rich protein-1; PfPTP1 = PfEMP1 trafficking protein-1; PTP7 = PfEMP1 trafficking protein-7; HSP = heat shock protein.

**Table 2 tropicalmed-08-00353-t002:** Potential cytoadherence receptors.

Receptor	Description
CD36 [94]	A scavenger receptor [95] expressed on the surface of many cell types including endothelial cells in most tissues [96].
Endothelial protein C receptor [97]	A protein on the surface of endothelial cells that binds to protein C and regulates thrombosis and inflammation [98].
Intercellular adhesion molecule-1 [99]	A member of the immunoglobulin-like superfamily that is expressed on the surface of many cell types and that is upregulated during inflammation [100].
Vascular cell adhesion molecule-1 [101]	A member of the immunoglobulin-like superfamily expressed on endothelial cells that plays a role in the adhesion of leukocytes [102].
Platelet endothelial cell adhesion molecule-1 [103]	A member of the immunoglobulin-like superfamily that is expressed on endothelial cells as well as platelets, monocytes, and granulocytes.
Neural cell adhesion molecule [104]	A member of the immunoglobulin-like superfamily that plays a wide range of roles in cellular adhesion [105].
Endothelial leukocyte adhesion molecule-1 [101]	A selectin, also known as E-selectin, expressed on endothelial cells that mediates adherence of leukocytes during inflammation [106].
P-selectin [107]	An adhesion protein expressed on the surface of activated platelets and endothelial cells during inflammation [106].
Thrombospondin [108]	A glycoprotein that is released into the blood plasma following thrombin-mediated activation of platelets that may function as a bridging molecule between infected erythrocytes and endothelial cells.
α_V_β_3_ integrin [109]	A member of a large family of adhesins that facilitate cell–cell interactions and interactions with the extracellular matrix [110].
Fibronectin [111]	An extracellular matrix protein. Binding may be related to the Arg–Gly–Asp (RGD) sequence present in thrombospondin and other serum proteins.
Complement receptor-1 [112]	A protein on the surface of erythrocytes and other cells that binds to complement proteins C3b and C4b and inhibits complement activation [113].
gC1qR/HABP1/p32 [114]	A 32 kDa protein (p32) that binds multiple proteins including complement C1q, hyaluronic acid (HA), and several other proteins [115].
Fractalkine [116]	A membrane-bound chemokine expressed on the surface of endothelial cells and associated with inflammation.
Chondroitin sulfate A [117,118]	A glycosaminoglycan that is secreted by placental cells and that lines the intervillous space that makes up the maternal–fetal interface.
Hyaluronic acid [119]	Another glycosaminoglycan of the placental intervillous space.
Heparin sulfate [120]	A glycosaminoglycan found on many cell types that may participate in rosetting.
Type A or B blood group antigens [121]	Trisaccharides on glycoproteins and glycolipids of the erythrocyte surface that determine ABO blood types and that may participate in rosetting.

**Table 3 tropicalmed-08-00353-t003:** Receptor binding specificity of PfEMP1 variants.

Receptor	PfEMP1 Domain(s) That Bind Receptor
CD36	DBLα/CIDRα head group [78], CIDRα2–6 [123,124]
Endothelial protein C receptor	CIDRα1 associated with DC8 or DC13 [97]
Intercellular adhesion molecule-1	DBLβ [125,126,127], DBLβ3 in DC4 [128]
Platelet endothelial cell adhesion molecule-1	DBLα/CIDRα head group or DBLδ2 [78], DC5 [129]
gC1qR	DBLβ12 associated with DC8 [130]
Complement receptor-1	DBLα [112]
Heparin sulfate	DBLα/CIDRα head group [78], DBLα1 [131]
Blood group antigens	DBLα/CIDRα head group [78]
Chondroitin sulfate A	First four modules including three unique DBLs and unique CIDR in var2csa [132], second unique DBL in var2csa [133]

**Table 4 tropicalmed-08-00353-t004:** Manifestations of severe malaria and indicators of poor prognosis ^a^.

Manifestation	Features
Severe anemia	Primarily in young children and defined as hematocrit < 15% or hemoglobin < 50 g/L in the presence of parasitemia.
Cerebral malaria	An unrousable coma in the presence of parasitemia and not attributable to another cause.
Respiratory distress	Defined by labored breathing and pulmonary edema that can progress to an acute respiratory distress syndrome requiring mechanical ventilation.
Impaired consciousness	An impaired consciousness that is less pronounced than the unrousable coma associated with cerebral malaria.
Prostration or weakness	Patients are unable to sit or walk, which is not attributable to neurological or other explanations.
Convulsions	Three or more repeated generalized convulsions observed within 24 h.
Acidosis	An important cause of death due to the accumulation of organic acids, including lactic acid, and compounded by ketoacidosis and acute kidney injury.
Hypoglycemia	Results from increased glucose consumption in the tissues and impaired hepatic glucogenesis. Often concomitant with lactic acidosis.
Jaundice	Results from a combination of hemolysis and hepatocyte damage and defined by elevated serum bilirubin in the presence of parasitemia.
Renal impairment	Defined by low urine output and high serum creatinine or urea levels despite adequate hydration.
Abnormal bleeding	Recurrent or prolonged bleeding from nose, gums, or venipuncture sites.
Coagulopathy	Activation of blood coagulation including disseminated intravascular coagulation or depletion of platelets.
Circulatory collapse (shock)	Defined as systolic blood pressure < 70 mm Hg in malaria patients and accompanied by cold clammy skin.
Hyperpyrexia	Core body temperature > 40 °C and may be associated with rapid heart rate and occasionally delirium.
Hyperparasitemia	Poor prognosis associated with >10% parasitized erythrocytes.

^a^ Based on clinical descriptions of the World Health Organization [150].

## Data Availability

Not applicable.

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
