# Peer review of "Knobs, Adhesion, and Severe Falciparum Malaria"

_tropicalmed, 2023, doi:10.3390/tropicalmed8070353_

Round 1

Reviewer 1 Report

The article addresses an important topic of great interest

The title of the article is consistent with the topic developed

The author is a specialist on the subject and he made an excellent review including a large number of important references on the topic that he developed.

The article is well structured and easy to understand.

The author included relevant and current references

The images are of good quality and illustrative.

In the tables, the author made a good summary with relevant references.

Author Response

Thank you!

Reviewer 2 Report

Dear Author(s)

The review article that has been obtained by you is an informative and useful literature, except:

1- It is obvious that just some of the knob-containing Plasmodium falciparum strains develop into complicated form of falciparum malaria infection.The logic of such phenomenon is not clear in your review. Clear it please.

2- The main reason of  organ dysfunction of black water fever syndrome( as a result of renal impairment) is not clear in your review. Give some more explanations about that please. 

Author Response

In regards to point 1, a paragraph has been added describing other factors involved in the development of severe falciparum malaria. 

In regards to point 2, a statement about kidney impairment and a reference has been added. I avoided the term 'black water fever' since there are different uses of this term and there is some confusion. For example, one definition is specific reference to blood in the urine associated with treatment of acute malaria. Treatment is not covered in this review. 

Reviewer 3 Report

This is an excellent comprehensive review on P. falciparum knobs, cytoadherence, and the resulting severe falciparum malaria. All tables and figures are very helpful for a better understanding of the complex interaction between the parasite, the human host, and severe clinical manifestation.

The summary at the end of this review is quite similar to the abstract. It would have been more interesting had the author provided a concluding paragraph with future perspectives. Because of the very many abbreviations used in the text, a list of abbreviations added at the end of the paper (before the list of references) may be helpful for readers.

The paper is very well written.

MAJOR COMMENTS:

none

MINOR COMMENTS:

Line 4: delete “affiliation 1”

Line 19: delete the extra space after “chronic infections.”

Lines 32-40, erythrocytic cycle: Please add the duration of one complete cycle and maybe also the approximate duration of the ring stage, trophozoite stage, and schizont stage. This information would be helpful later for uninitiated readers, in line 178 (“approximately halfway through the 48-hour replicative cycle”).

Fig. 3: I would move the vertical arrow pointing at the intron a few millimeters to the left. It may be helpful to add (in parentheses) the approximate size of the intron in line 139: “The gene contains a single intron (XX bp) that is located...”

Line 144, Fig. 3 legend: Please delete the extra period.  .

Line 157: heat shock protein 40 (HSP40)

Line 159, “PFE1605w binds the ATS”: Please check the meaning. Does it bind ATS or does it bind to ATS?

Line 168: 16 hours (as written in line 174)

Line 225: Delete the extra space at the beginning of the sentence “A protein called ankyrin binds to...

Fig. 4 legend: For further clarity, please specify what each structure represents on the left-hand side of the figure: (1) light blue oval structures (band 4.1R protein?), (2) green oval structure (ankyrin?), (3) dark blue oval structures (KAHRP? as in the right-hand side figure).   

Line 242: ankyrin suggesting that KAHRP...

Line 246, “it has been suggest”: It has been suggested...

Line 267, “Maurer’s clefts with its PfEMP1 cargo”: Maurer’s clefts with their PfEMP1 cargo

Line 296, “...involved in strained dependent adherence...”: strain-dependent adherence

Lines 297-299, “Cloning and sequencing studies revealed the PfEMP1 was a member...”: revealed that PfEMP1 is (?) a member...

Lines 311 and 314, “ATS, DBL, CIDR”: These abbreviations have already been introduced earlier in the text (fig 3 legend). The author does not need to repeat what these abbreviations stand for again.

Lines 360-361, “The expressed PfEMP1 allele, along with..., are localized”: The subject of the sentence is singular (PfEMP1 allele) (whatever follows “along with” is not part of the subject); therefore, is localized...unless the author prefers to say “The expressed PfEMP1 allele and the appropriately modified histones...are localized”

Line 366, “modificatioins”: modifications (spelling)

Lines 407-408, “the large number of potential receptors and the strain specificity of receptor binding was...”: What is the subject of this sentence? If it’s “the large number...and the strain specificity (plural)” then the verb is “were.”

Line 404, Table 2, “chondroitin sulphate A” and Line 446, Table 3, last row, “chondroitin sulphate A”: The author has been using American spelling up to now. It should be spelled “chondroitin sulfate A” in both tables.

Line 452, “bind ICAM-1”: bind to (?) ICAM-1

Line 456, “A single PfEMP1 allele binds to chondroitin sulphate A in the placenta”: sulfate

Line 457: sulfate

Line 461, “heparan sulphate”: heparin sulfate

Lines 466-471, “var2csa”: The author is referring to the allele (gene), which, according to the international nomenclature, should be italicized.

Line 491, “clinical manifestations...range from asymptomatic carriers to a febrile illness to severe organ dysfunction”: “Asymptomatic carriers” is not a clinical manifestation, but persons. I suggest “asymptomatic carriage” instead so that the series cited here (asymptomatic carriage, febrile illness, severe organ dysfunction) is more coherent and of the same nature.

Line 502, Table 4: This table needs a reference citation. The definitions of severe malaria have been changing, and the manifestations are usually different in children and adults. I think that the latest (2014) WHO definition should be cited here, and the table revised accordingly. For example, the author does not mention “bleeding” or “disseminated intravascular coagulation.” Please see Table 4 Epidemiological and research definition of severe falciparum malaria in: World Health Organization. Severe Malaria. Trop Med Int Health. 2014;19(Suppl 1):1–131. doi: 10.1111/tmi.12313_2.

Lines 516-518, “hemozoin causes global activation of lung epithelial cells and promotes lung inflammation”: This is not a clinical observation, and not even an in vitro experimental finding. The authors of REF 151 showed epithelial cell activation using RNA sequencing. These lines seem to go way beyond the clinical description of respiratory distress described in this paragraph (lines 512-518) and does not seem to support what the author describes in lines 512-516.

Lines 521-525, “uninfected erythrocytes are destroyed at higher rates...due to complement-mediated lysis and phagocytosis mediated by immune complex deposition or complement activation. Furthermore,...decreased production of erythrocytes...”: Please add literature citations unless REF 152 covers these mechanisms of severe anemia in malaria.

Lines 536-537, “decreased perfusion combined with parasite metabolism results in hypoglycemia and acidosis”: The actual mechanisms leading to hypoglycemia (and metabolic acidosis) are not well known. In addition to the fact that malaria parasites consume glucose (this may be an important factor to consider in case of hyperparasitemia), some authors argue that fasting (or loss of appetite) should be taken into consideration, while others hypothesize that cytokines may impair gluconeogenesis.

Line 578, REF 160: It seems that REF 160 was not cited in the text. Please check.

Line 579, “bind both EPCR and ICAM1”: bind to?

Line 583, “heparan sulphate”: heparin sulfate

Lines 585-596, “var2csa”: As in lines 466-471, the author is referring to the allele (gene), which, according to the international nomenclature, should be italicized.

Several references need minor corrections:

REF 1, 30, 51, 82, 88, 106, 110, 113, 136, 139: please check the format of “doi” and use the same format throughtout.

REF 6: The article number is missing: Warncke JD, Beck HP. Host cytoskeleton remodeling throughout the blood stages of Plasmodium falciparum. Microbiol Mol Biol Rev. 2019, 83, e00013-19. doi: 10.1128/MMBR.00013-19.

REF 35 and REF 61: journal name: Front. Cell (capital letter “C”) Dev. Biol.

REF 64: The volume number is missing: Jonsdottir TK, Counihan NA, Modak JK, Kouskousis B, Sanders PR, Gabriela M, Bullen HE, Crabb BS, de Koning-Ward TF, Gilson PR. Characterisation of complexes formed by parasite proteins exported into the host cell compartment of Plasmodium falciparum infected red blood cells. Cell Microbiol. 2021, 23, e13332. doi: 10.1111/cmi.13332.

REF 68: “doi”: Udeinya IJ, Miller LH, McGregor IA, Jensen JB. Plasmodium falciparum strain-specific antibody blocks binding of infected erythrocytes to amelanotic melanoma cells. Nature. 1983, 303, 429-431. doi: 10.1038/303429a0.

REF 70: “doi”: Howard RJ, Barnwell JW, Rock EP, Neequaye J, Ofori-Adjei D, Maloy WL, Lyon JA, Saul A. Two approximately 300 kilodalton Plasmodium falciparum proteins at the surface membrane of infected erythrocytes. Mol Biochem Parasitol. 1988, 27, 207-223. doi: 10.1016/0166-6851(88)90040-0.

REF 71: Plasmodium falciparum (in italics)

REF 72: The article number is missing: Otto TD, Böhme U, Sanders M, Reid A, Bruske EI, Duffy CW, Bull PC, Pearson RD, Abdi A, Dimonte S, Stewart LB, Campino S, Kekre M, Hamilton WL, Claessens A, Volkman SK, Ndiaye D, Amambua-Ngwa A, Diakite M, Fairhurst RM, Conway DJ, Franck M, Newbold CI, Berriman M. Long read assemblies of geographically dispersed Plasmodium falciparum isolates reveal highly structured subtelomeres. Wellcome Open Res. 2018, 3, 52. doi: 10.12688/wellcomeopenres.14571.1. In addition, “[version 1; peer review: 3 approved]” can be deleted.

REF 73: “doi” is missing: Claessens A, Hamilton WL, Kekre M, Otto TD, Faizullabhoy A, Rayner JC, Kwiatkowski D. Generation of antigenic diversity in Plasmodium falciparum by structured rearrangement of Var genes during mitosis. PLoS Genet. 2014, 10, e1004812. doi: 10.1371/journal.pgen.1004812.

REF 74: The journal name is incomplete (USA) and “doi” is missing. Also, please correct the third author’s name (Wellems, T.E.): Peterson DS, Miller LH, Wellems TE. Isolation of multiple sequences from the Plasmodium falciparum genome that encode conserved domains homologous to those in erythrocyte-binding proteins. Proc Natl Acad Sci U S A. 1995, 92, 7100-7104. doi: 10.1073/pnas.92.15.7100.

REF 76: Please correct the journal name: Baruch DI, Ma XC, Singh HB, Bi X, Pasloske BL, Howard RJ. Identification of a region of PfEMP1 that mediates adherence of Plasmodium falciparum infected erythrocytes to CD36: conserved function with variant sequence. Blood. 1997, 90, 3766-3775.

REF 77: Please add the “doi”: Rask TS, Hansen DA, Theander TG, Gorm Pedersen A, Lavstsen T. Plasmodium falciparum erythrocyte membrane protein 1 diversity in seven genomes--divide and conquer. PLoS Comput Biol. 2010, 6, e1000933. doi: 10.1371/journal.pcbi.1000933.

REF 122: The article number is missing: Bachmann A, Metwally NG, Allweier J, Cronshagen J, Del Pilar Martinez Tauler M, Murk A, Roth LK, Torabi H, Wu Y, Gutsmann T, Bruchhaus I. CD36 - A host receptor necessary for malaria parasites to establish and maintain infection. Microorganisms. 2022, 10, 2356. doi: 10.3390/microorganisms10122356.

REF 127: “doi”: Oleinikov AV, Amos E, Frye IT, Rossnagle E, Mutabingwa TK, Fried M, Duffy PE. High throughput functional assays of the variant antigen PfEMP1 reveal a single domain in the 3D7 Plasmodium falciparum genome that binds ICAM1 with high affinity and is targeted by naturally acquired neutralizing antibodies. PLoS Pathog. 2009, 5, e1000386. doi: 10.1371/journal.ppat.1000386.

REF 134: Please add the article number: Avril M, Bernabeu M, Benjamin M, Brazier AJ, Smith JD. Interaction between endothelial protein C receptor and intercellular adhesion molecule 1 to mediate binding of Plasmodium falciparum-infected erythrocytes to endothelial cells. mBio. 2016, 7, e00615-16. doi: 10.1128/mBio.00615-16.

REF 140: “doi”: Viebig NK, Levin E, Dechavanne S, Rogerson SJ, Gysin J, Smith JD, Scherf A, Gamain B. Disruption of var2csa gene impairs placental malaria associated adhesion phenotype. PLoS One. 2007, 2, e910. doi: 10.1371/journal.pone.0000910.

REF 142: Please add the article number: Chaudhary A, Kataria P, Surela N, Das J. Pathophysiology of cerebral malaria: implications of MSCs as a regenerative medicinal tool. Bioengineering (Basel). 2022, 9, 263. doi: 10.3390/bioengineering9060263.

REF 163: Please add the article number: Storm J, Jespersen JS, Seydel KB, Szestak T, Mbewe M, Chisala NV, Phula P, Wang CW, Taylor TE, Moxon CA, Lavstsen T, Craig AG. Cerebral malaria is associated with differential cytoadherence to brain endothelial cells. EMBO Mol Med. 2019, 11, e9164. doi: 10.15252/emmm.201809164.

REF 166: Please add the article number: Joste V, Guillochon E, Fraering J, Vianou B, Watier L, Jafari-Guemouri S, Cot M, Houzé S, Aubouy A, Faucher JF, Argy N, Bertin GI. PfEMP1 A-type ICAM-1-binding domains are not associated with cerebral malaria in Beninese children. mBio. 2020, 11, e02103-20. doi: 10.1128/mBio.02103-20.

REF 168: “doi”: Mayor A, Hafiz A, Bassat Q, Rovira-Vallbona E, Sanz S, Machevo S, Aguilar R, Cisteró P, Sigaúque B, Menéndez C, Alonso PL, Chitnis CE. Association of severe malaria outcomes with platelet-mediated clumping and adhesion to a novel host receptor. PLoS One. 2011, 6, e19422. doi: 10.1371/journal.pone.0019422.

Please see my comments to the author.

Author Response

I thank reviewer 3. It was good to get another set of eyes to find all of the minor errors. That was a big help and I am grateful for the time spent. 

As suggested I added some future perspectives to the summary and added a list of abbreviations. 

In regards to the minor comments and corrections to the references, I accepted all of the corrections and suggestions and made changes to the manuscript accordingly. I also confirmed that reference 152 also extensively discussed destruction of erythrocytes.